# Oxygen Transfer Capacity as a Measure of Water Aeration by Floating Reed Plants: Initial Laboratory Studies

**Antonio Albuquerque** [1], **Peter Randerson** [2,*] **and Andrzej Białowiec** [3]

1   Department of Civil Engineering and Architecture and FibEnTech, University of Beira Interior, Edificio 2 das Engenharias, Calcada Fonte do Lameiro, 6201-001 Covilha, Portugal; antonio.albuquerque@ubi.pt

2   School of Biosciences, Cardiff University, Cardiff CF10 3AX, UK

3   Institute of Agricultural Engineering, Faculty of Life Sciences and Technology, Wroclaw University of Environmental and Life Sciences, 37a Chełmońskiego Str., 51-630 Wroclaw, Poland; andrzej.bialowiec@upwr.edu.pl

*   Correspondence: randerson@cardiff.ac.uk; Tel.: +44-29-2087-4148

**Abstract:** Reed-*Phragmites australis* (Cav.) Trin. ex Steud, an aquatic plant, commonly used in constructed wetlands for wastewater treatment, supplies oxygen into the subsurface environment. Reed may be used as a 'green machine' in the form of a floating vegetation cover with many applications: wastewater lagoons, manure lagoons or sewage sludge lagoons. An important measure of the performance of the plant system is the oxygen transfer capacity (OTC). Accurate prediction of the OTC in relation to reed biomass would be crucial in modelling its influence on organic matter degradation and ammonia–nitrogen oxygenation in such lagoons. Laboratory experiments aiming to determine OTC and its dependence on reed biomass were carried out. Eight plants with a total dry mass ranging from approximately 3 to 7 g were tested. Mean OTC was determined per plant: $0.18 \pm 0.21$ (g $O_2 \cdot m^{-3} \cdot h^{-1} \cdot plant^{-1}$), with respect to leaves-and-stem dry mass (dlsm): $44.91 \pm 35.21$ (g $O_2 \cdot m^{-3} \cdot h^{-1} \cdot g$ dlsm$^{-1}$), and to total dry mass (dtm): $33.25 \pm 27.97$ (g $O_2 \cdot m^{-3} \cdot h^{-1} \cdot g$ dtm$^{-1}$). In relation to the relatively small root dry mass (drm), the OTC value was $136.02 \pm 147.19$ (g $O_2 \cdot m^{-3} \cdot h^{-1} \cdot g$ drm$^{-1}$). Measured OTC values varied widely between the individual plants (variation coefficient 115%), in accordance with their differing size. Oxygenation performance was greatest in the reed plants with larger above ground dry mass (>4 g dlsm), but no influence of the root dry mass on the OTC rate was found.

**Keywords:** oxygen transfer capacity; reed; rhizosphere; oxygen release; *Phragmites australis* (Cav.) Trin. ex Steud; constructed wetlands; reed floating cover

## 1. Introduction

Problematic by-products of wastewater, manure and sewage sludge storage in lagoons are emissions of odour, volatile organic compounds (VOCs), ammonia ($NH_3$), hydrogen sulphide ($H_2S$) and greenhouse gases (GHGs) ($CH_4$, $N_2O$, and $CO_2$). Local and regional air quality and potential contribution to climate change [1] due to these emissions require the need for mitigation technologies that target multiple emissions and are cost-efficient to put into practice [2]. In rural areas, especially in developing countries, where technical infrastructure development is weak, extensive solutions based on natural processes are applicable. One solution is to create floating emergent macrophyte treatment wetlands, a hybrid of ponds and wetlands, that offer potential advantages for the treatment of polluted waters with highly variable flows. A plant commonly used in both rooted and floating systems is reed *Phragmites australis* (Cav.) Trin. ex Steud [3].

An important measure of the performance of reed floating cover (RFC) is the oxygen transfer capacity (OTC). According to Wießner et al. [4] oxygen transport results in the formation of an aerobic layer around the roots of a thickness dependent on the oxidation–reduction (redox) status of the rhizosphere. Within this layer are redox gradients with values (Eh) from +500 mV (directly adjacent to the root surface) to −200 mV (away from the roots), a picture that is generally characteristic of rhizospheric conditions [5]. Wießner et al. [4] found that the rate of $O_2$ release in the rhizosphere can be affected by redox potential, with −250 to −150 mV being the optimum values for $O_2$ release in some plant species. Low initial values of redox potential seemed to stimulate plants to supply $O_2$ intensively, causing an increase. When redox potential values stabilized, plants decreased the release of $O_2$. Gradients of dissolved oxygen (DO) and oxidation–reduction potential (ORP) associated with the boundaries of aerobic zones have been measured using micro-electrodes in soil and wetland root zones [6,7]. Such micro-gradients indicate the presence of distinct microbial processes, as well as particular dissolved gases within the substratum matrix and the rhizosphere zone surrounding plant roots. Rhizosphere oxygenation by radial oxygen loss (ROL) from roots is of great importance for wetland plants to overcome anaerobic conditions. ROL from some plants has been shown to increase soil Eh [4,6,8–10], which enables those plants to survive in otherwise anoxic conditions.

Most aquatic macrophyte species have aerenchyma tissue, which enables the transfer of atmospheric oxygen via stems to roots, much of which then leaks into the soil substrate [6,11]. It was shown by Albert et al. [12] for *Schoenoplectus pungens* (Vahl) Palla (bulrush), that aerenchyma tissue can increase with flooding, that stem flexibility possibly reduced, and stem diameter and height increased as water depths or flooding increased in aquatic macrophytes. Li et al. [13] showed that the development of aerenchyma tissue in willow (*Salix nigra*) depends on soil water conditions. After 28 days of treatment, the root porosity of plants growing in an aerated zone was 33%, plants growing in soil with full water saturation was 28.6%, but plants growing in soil partially flooded had root porosity 39.2%. The oxygen release appears to be dependent on photosynthetic activity and environmental factors. Oxygen release by the roots of woody plants has been shown in *Alnus glutinosa* Gaertn. [14], *Taxodium distichum*, *Betula pubescens* Ehrh., and *Populus tremulus* L. [15], *A. japonica* (Thunb.) Steud., *A. hirsuta* (Spach) Rupr [16], *Salix alba* L., *S. cinerea* and *S. viminalis* L. [17], *S. viminalis* L. [6], *S. nigra* [13].

Predictions of the oxygen release from the roots, the amount that might be supplied to the soil, and hence the extent of oxidized rhizospheres have already been determined for *Phragmites australis* (Cav.) Trin. ex Steud [8,18], but only in relation to 2-dimensional soil porous area of horizontal or vertical subsurface constructed wetlands. In the case of RFC, a 3-D approach is required to simulate the amount of oxygen available to the whole volume of the pond. To date, the 3-D OTC in relation to the plant biomass of RFC has not been determined. OTC defines the amount of oxygen introduced into the water volume within a unit of time and is expressed as mg $O_2$ $L^{-1} \cdot h^{-1}$ [6].

The oxygen transfer capacity (OTC) of *Phragmites australis* (Cav.) Trin. ex Steud per plant (or per plant mass) is not well known, but it would be a valuable tool in predicting and modelling RFC performance. Optimizing the efficiency of phytoremediation in lagoons covered by floating mats of reed is important, as the technology of floating covers of reed and other plants is developing and being applied in different water treatment solutions [19].

The aim of this study was to determine the OTC of suspended reed plants and to estimate the influence of reed biomass on OTC value.

## 2. Materials and Methods

### 2.1. Experimental Set-Up

The experimental set-up (Figure 1) included a 1500 mL beaker (13.5 cm in diameter), with 1000 mL of deionized water, previously sparged with nitrogen gas for 20 min to remove the dissolved oxygen (DO). The endpoint of sparging was achieved when oxygen was below 5% saturation, measured with an oxygen sensor CellOx 325, connected to a Multi 340i meter (WTW, Germany). A

single reed plant (*Phragmites australis* (Cav.) Trin. ex Steud) was suspended in the beaker, together with three microelectrodes, installed close to the root zone, to measure DO (sensor OX-N, 1 mm), oxidation–reduction potential (ORP) (sensor RD-N, 1 mm) and reaction pH (sensor pH-N, 1 mm) throughout the experiment (sensors by Unisense, Denmark). The OX-N microelectrode was connected to an Oxy-meter (4-channel picoammeter PA2000, Unisense, Denmark) and the RD-N and pH-N microelectrodes were connected to a 4-channel PHM2010 millivoltmeter (Unisense, Denmark). Water temperature was also measured through a Sentix41 electrode connected to a Multiline 320i meter (WTW, Germany). A one cm layer of paraffin oil was placed on top of the liquid to prevent oxygen diffusion to water from the air. Water in the beaker and through the root zone was mixed by a magnetic stirrer at the base (100 rpm, Nahita, UK) without disturbing the impervious surface layer. The lack of disturbance to this essential barrier was determined experimentally, by observing the system while applying different rpm. The room air temperature and the water temperature in the beaker were ~20 °C with natural light.

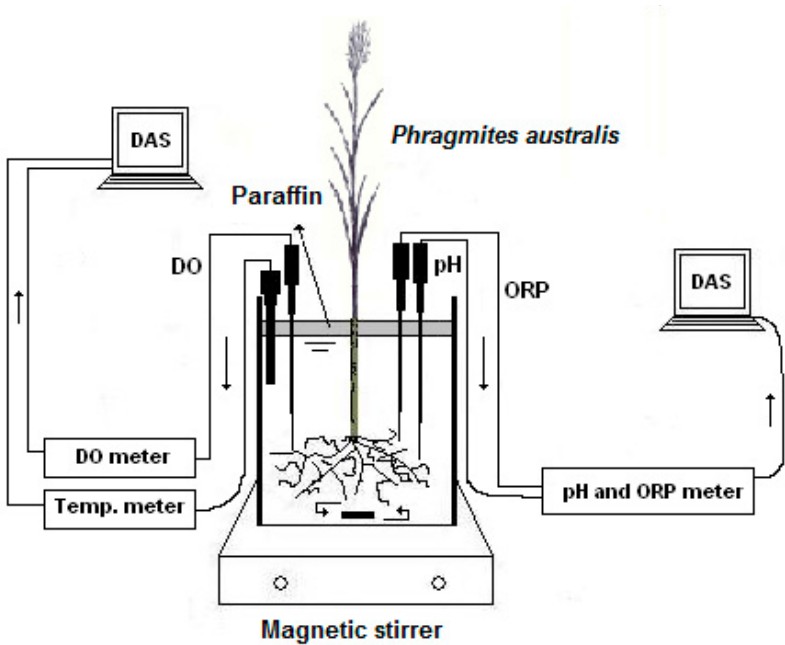

**Figure 1.** Schematic representation of the experimental set-up (DAS: data acquisition systems; ORP: oxidation-reduction potential; DO dissolved oxygen).

*2.2. Experimental Procedure*

Reed plants were taken from a constructed wetland mesocosm, described in Albuquerque et al. [20] during the spring–summer period. Individual plants were carefully removed from the media substratum, rhizomes were flushed with clean water, and dead rhizomes were removed. Sample plants numbered 1–8, selected for a wide range of size, were used to assess the OTC (Table 1). The average duration of each experiment was 2.35 h. At the end of each experiment, the plant was dried at 105 °C in a dry oven (JP Selecta 2000200, Spain) and weighed to determine the dry mass (roots (drm); leaves-and-stems (dlsm); and total dry mass (dtm)). DO was measured continuously during all experiments but connected to data acquisition systems (DAS) programmed to store measurements only each 1 min.

**Table 1.** Measured oxygen transfer capacity and morphometric parameters of individual plants used in the experiment.

| Plant Number | Oxygen Transfer Capacity | | | | Root Dry Mass (drm) | Leaves and Stem Dry Mass (dlsm) | Total Plant Dry Mass (dtm) |
|---|---|---|---|---|---|---|---|
| | (g $O_2 \cdot m^{-3} \cdot h^{-1} \cdot plant^{-1}$) | (g $O_2 \cdot m^{-3} \cdot h^{-1} \cdot g\ drm^{-1}$) | (g $O_2 \cdot m^{-3} \cdot h^{-1} \cdot g\ dlsm^{-1}$) | (g $O_2 \cdot m^{-3} \cdot h^{-1} \cdot g\ dtm^{-1}$) | (g) | (g) | (g) |
| 1 | 0.65 | 477.49 | 112.55 | 91.08 | 1.36 | 5.77 | 7.13 |
| 2 | 0.10 | 122.68 | 46.01 | 33.46 | 0.81 | 2.16 | 2.97 |
| 3 | 0.08 | 95.63 | 28.99 | 22.24 | 0.87 | 2.87 | 3.74 |
| 4 | 0.03 | 33.66 | 14.03 | 9.90 | 1.03 | 2.47 | 3.50 |
| 5 | 0.32 | 169.39 | 78.53 | 53.65 | 1.91 | 4.12 | 6.03 |
| 6 | 0.14 | 121.74 | 46.67 | 33.73 | 1.15 | 3.00 | 4.15 |
| 7 | 0.07 | 50.39 | 23.94 | 16.23 | 1.33 | 2.80 | 4.13 |
| 8 | 0.03 | 17.17 | 8.56 | 5.71 | 1.75 | 3.51 | 5.26 |
| Mean | 0.18 | 136.02 | 44.91 | 33.25 | 1.28 | 3.34 | 4.61 |
| Standard deviation | 0.21 | 147.19 | 35.21 | 27.97 | 0.40 | 1.15 | 1.41 |
| Variation coefficient | 119.62 | 108.22 | 78.40 | 84.11 | 30.99 | 34.57 | 30.55 |

## 2.3. Oxygen Transfer Capacity (OTC) Determination

OTC (mg $O_2$ $L^{-1} \cdot h^{-1}$) was calculated using Equations (1) to (4) used by Randerson et al. [6]:

$$OTC = 11.33 \cdot \frac{1}{\Delta T} \cdot ln\frac{D_0}{D_t} \cdot \sqrt{\frac{k_{10}}{k_t}} \tag{1}$$

where 11.33 is the maximum oxygen saturation (mg $L^{-1}$) in water at 10 °C and atmospheric pressure (1013 hPa), $\Delta T$ the duration of the measurements (h), $D_0$ the initial oxygen deficit ($C_S$–$C_0$) (mg $L^{-1}$), Dt the oxygen deficit at time t ($C_S$–Ct) (mg $L^{-1}$), $C_0$ the initial oxygen concentration (mg $L^{-1}$), $C_S$ the maximum oxygen concentration (mg $L^{-1}$), Ct the oxygen concentration at time t (mg $L^{-1}$) and $K_{10}/K_t$ the coefficient for temperature compensation for temperature 20 °C (0.784, according to Kowalska [21]).

The $\Delta T$ may be calculated from Equation (2) and Equation (3), where tan $\alpha$ angle is the rate of oxygenation:

$$f\left(log\frac{D_0}{D_t}\right) = \Delta T \tag{2}$$

$$tg\alpha = \frac{log\frac{D_0}{D_t}}{\Delta T} \tag{3}$$

Therefore, the OTC may be estimated through Equation (4):

$$OTC = 26.1 \cdot tg\alpha \cdot \sqrt{\frac{k_{10}}{k_t}} \tag{4}$$

The OTC shows the rate of oxygen release to water with the initial oxygen concentration, at a temperature of ~20 °C, and under atmospheric pressure.

The variability of OTC values is shown by the means, standard deviations, variation coefficients and determination coefficients ($R^2$), calculated using Statistica 12.0 (TIBCO Software Inc., Palo Alto, CA, USA).

## 3. Results and Discussion

The initial pH ranged between 6.58 and 6.81 (mean 6.69 ± 0.09). ORP values ranged from 15.50 to 184.30 mV (mean 108.24 ± 70.35 mV). The mean temperature for all experiments was 20.4 ± 0.4 °C. *Phragmites australis* is generally tolerant of temperature variations, but the optimum temperature for

growth is between 12 °C and 25 °C [22,23]. These values indicated quite stable conditions for plant activity [22,23] and oxidative conditions during the experiments.

The mean value of the OTC per plant (eight measurements) was $0.18 \pm 0.21$ (g $O_2 \cdot m^{-3} \cdot h^{-1} \cdot plant^{-1}$) (mean $\pm$ SD) (Table 1). The values of the OTC (g $O_2$ $m^{-3} \cdot h^{-1} \cdot plant^{-1}$) varied widely between individual plants (maximum OTC 0.65 for plant 1; minimum 0.03 for plants 4, 8; variation coefficient 119%) because of the variation in the size of the selected individuals. A similar value of OTC of 0.18 (g $O_2$ $m^{-3} \cdot h^{-1} \cdot plant^{-1}$) for willow had been obtained by Randerson et al. [6], but in that case the variability between willow plants was smaller (standard deviation 0.07 (g $O_2$ $m^{-3} \cdot h^{-1} \cdot plant^{-1}$)). Willows used in that experiment were of the same age and similar sizes and they were clones of the same plant. In the present study, plant morphometric parameters also varied widely (variation coefficients between 31 and 34%).

For phytoremediation, it is typical to express the rate of oxygen release based on the biomass per unit area of shoots and the associated roots per $m^2$ per day. In the current experiment, there was a single shoot and associated roots within a volume of water medium, giving a mean oxygen release rate of $0.18 \pm 0.21$ g $O_2 \cdot m^{-3} \cdot h^{-1} \cdot plant^{-1}$. Assuming oxygen is released into the upper 0.1 m layer of the root zone, with 45 shoots per $m^2$ (a mean value for reed growing as floating cover [20]), and with 1 $m^3$ volume having a surface area of 1 $m^2$, then the 2-D rate of oxygen release would be $45 \times 0.18 \times 0.124$, giving a mean value of 19.4 g$\cdot O_2 \cdot m^{-2} \cdot d^{-1}$. Wießner et al. [4] estimated ca 4 g$\cdot O_2 \cdot m^{-2} \cdot d^{-1}$ for *Typha latifolia* and for *Phragmites*, and Armstrong and Armstrong [9] estimated an oxygen release of 5–12 g$\cdot O_2 \cdot m^{-2} \cdot d^{-1}$. The obtained value was higher than the values from earlier field studies but of the same order of magnitude.

The present study under laboratory conditions showed that, in relation to root dry mass (drm), the reed supplied oxygen at an average rate of 136 (g $O_2 \cdot m^{-3} \cdot h^{-1} \cdot kg_{drm}^{-1}$). In previous work on willows [6], the OTC related to wet root mass (rwm), was 196 (g $O_2 \cdot m^{-3} \cdot h^{-1} \cdot kg_{rwm}^{-1}$). Assuming the reed root moisture to be 50%, the OTC in relation to root wet mass should be at a level of 68 (g $O_2 \cdot m^{-3} \cdot h^{-1} \cdot kg_{rwm}^{-1}$), which is about 1/3 of the values determined for willow. The average reed OTC value in relation to the total dry mass was 33.25 (g $O_2 \cdot m^{-3} \cdot h^{-1} \cdot kg_{dtm}^{-1}$) (Table 1).

Values of the OTC clearly increased in the largest reed plants measured, having a dry mass of leaves-and-stems in excess of 4 g (Figure 2B). The same trend was apparent with the reed total dry mass (Figure 2C) but not with dry root mass (Figure 2A), because the dry root mass was relatively small in the plants studied and varied little between them. These results indicated that plants with larger above-ground parts, with a greater area for light absorption and assimilation, are more effective in releasing oxygen from the roots into the surrounding water. A similar phenomenon was observed by Randerson et al. [6] in experiments with willows, although the OTC showed a positive relationship with the both root mass and leaf area. In the present study, the relationships with dry mass were illustrated by Equation (5):

$$OTC = a_1 + e^{(-a_2 + a_3 \cdot x)} \tag{5}$$

where $a_1$, $a_2$, $a_3$ are equation parameters, x is dslm or dtm.

In the case of willows [6], the positive relationships between the OTC and both the root mass and leaf area were described by an asymptotic equation (Equation (6)):

$$OTC = a_1 + \left(1 - e^{(-a_2 \cdot (x - a_3))}\right) \tag{6}$$

Equation (6) indicates an upper limit for OTC with increasing willow mass whereas for reed, Equation (5) may apply only to relatively small, young plants. The observed lack of influence of root mass on OTC could be related to the release of root exudates [6] into the aquatic environment, supplying organic compounds to the microbial biofilm and leading to oxygen depletion. In our study, an exponential curve (Equation (5); Figure 2B,C) was defined by eight size-selected plants, whilst other studies used similarly small samples: four shoots of reed per treatment [7,9,13]; 4–10 replicates of

Salix [17]. Further investigations of OTC in larger, well developed reed plants and those with a larger root mass are needed to confirm the above relationships.

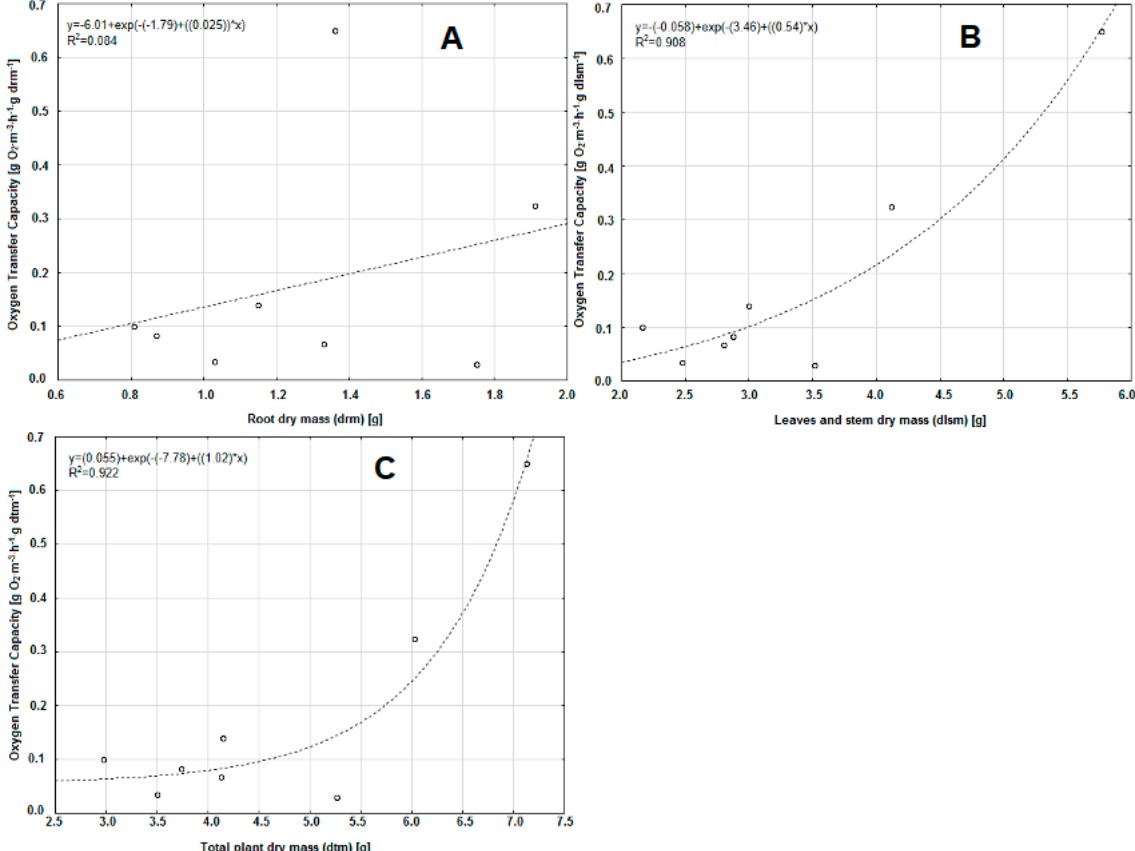

**Figure 2.** Relationships between the reed root dry mass (**A**); leaves and stem dry mass (**B**); total plant dry mass; (**C**) and oxygen transfer capacity (OTC).

It is well known that light intensity affects OTC in helophytes. For example, Williams et al. [11] showed large diurnal fluctuations of oxygen release from willow roots in a constructed wetland, and that the daily maximum rate was related to solar intensity (between sunny and cloudy days). In further studies with reed plants, light intensity should be included as an additional factor influencing OTC values, with a view to enabling the process of sub-surface oxygenation to be more comprehensively modelled.

## 4. Summary

The ability of reed to supply oxygen to the water environment has been experimentally proven in this work, confirming earlier qualitative results from laboratory and field systems shown by other authors. The oxygen transfer capacity (OTC) for individual reed plants was estimated at a mean level of $33.25 \pm 27.97$ per unit of total dry mass of plant (g $O_2 \cdot m^{-3} \cdot h^{-1} \cdot kg_{dtm}^{-1}$). Oxygenation performance, as measured by OTC, was greatest in plants with larger above ground dry mass (>4 g dlsm), but no influence of root dry mass on OTC rate was found. These results may have importance in modelling the prediction of redox conditions and changes in nitrogen and organic compounds in lagoons with reed floating covers. Further investigations concerning different phases of reed development, plant size, lighting conditions, and water–wastewater matrices used for reed cultivation are needed.

**Author Contributions:** Conceptualization, A.A., P.R. and A.B.; methodology, A.B.; validation, P.R. and A.B.; formal analysis, A.A., P.R.; investigation, A.A. and A.B.; resources, A.A.; data curation, P.R.; writing—original

draft preparation, A.A. and A.B.; writing—review and editing, A.B. and P.R.; visualization, A.B.; supervision, P.R. All authors have read and agreed to the published version of the manuscript.

**Funding:** This research received no external funding. The APC was funded by Cardiff University institutional Open Access fund.

**Acknowledgments:** This article is an output of the leading research team—Waste and Biomass Valorization Group (WBVG), Wroclaw University of Environmental and Life Sciences, Wroclaw, Poland. https://www.upwr.edu.pl/research/50121/waste_and_biomass_valorization_group_wbvg.html.

**Conflicts of Interest:** The authors declare no conflict of interest. The funders had no role in the design of the study; in the collection, analyses, or interpretation of data; in the writing of the manuscript; or in the decision to publish the results.

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
