# Peer review of "Oxygen Transfer Capacity as a Measure of Water Aeration by Floating Reed Plants: Initial Laboratory Studies"

_processes, doi:10.3390/pr8101270_

Round 1
Reviewer 1 Report
Dear authors, thanks for your manuscript. I believe that the topic is very interesting and it can be of high interest for readers.
Anyway, I believe that the methodology can be improved. In particular, the number of plants is not enough to detect a trend, based on the results you have.
High values of R2 you obtained are mainly due to the plant n.1 that has very high O2 transfer. If you try to delete this number the correlation goes down!
So, my suggestion is to use more than eight plants in order to verify the correlation before to publish the paper.
Reviewer 2 Report
The authors carried out a fine experiment to identify the O2 transer capacity of a very common wetland plant (reeds). It is an interesting work although in its initial study. I wish to suport it for publication. But I have the following comments:
1) The title is hard to follow, obviously it is the "Proof OF the concept:". It is also hard to me of the term of "reed floating cover". In addition, "initial study" at the end is also strange to read.
2) Only in abstract, Oxygen Transfer Capacity (OTC) was defined twice---an example of showing the careless of editing the paper.
3) "A 1cm of paraffin oil layer was placed on the top of the beaker, while the beaker was fully mixed during the test"----please explain it on how to keep the paraffin oil layer to be stable.
4) More importantly, it has been well accepted that the plant roots play the important role to transfer O2 to the wetland substrate/media for pollutants biodegradation via the biofilm. From this study there was no correlation between mass of roots and OTC. It should be well explained in the paper.
Round 2
Reviewer 1 Report
Dear author,
I still believe that the number of plants is not enough to have a valid correlation.
Round 3
Reviewer 1 Report
Dear authors,
thanks for your explanations. I hope you can verify your result with more plants in the future.